# Essential Trace Elements and Arsenic in Thermal Springs, Afghanistan

**Hussain Ali Jawadi** [1] , **Hasan Ali Malistani** [1], **Mohammad Anvar Moheghy** [1] **and Jay Sagin** [2,3,*]

1    Department of Geology, Faculty of Geosciences, Bamyan University, Bamyan City, Bamyan 1601, Afghanistan; hussainali.jawadi@daad-alumni.de (H.A.J.); hmalistani@gmail.com (H.A.M.); Moheghy@gmail.com (M.A.M.)
2    Department of Geosciences, Western Michigan University, Kalamazoo, MI 49008, USA
3    The Environment & Resource Efficiency Cluster (EREC), School of Engineering and Digital Sciences, Nazarbayev University Research and Innovation System, Nazarbayev University, Nur Sultan 010000, Kazakhstan
*    Correspondence: zhanay.sagintayev@nu.edu.kz

**Abstract:** Thermal springs are natural hydrogeological features which are highly affected by local volcanism or tectonic activity. Thermal springs are the best source of hydrothermal energy to heat houses and aid in the recovery of skin diseases. However, they consist of some heavy and trace metals such as arsenic, lead, zinc, copper, iron, and so forth. Somehow, the thermal springs of central Bamyan have become contaminated with some essential trace elements. Thus, this study was conducted to assess and determine the number of these trace elements in the thermal springs. To achieve these objectives, a preliminary survey, water sampling, and in situ measurements of physicochemical parameters were conducted in research areas. All the collected water samples were analyzed chemically to determine the amount of trace elements, including arsenic, barium, copper, iron, manganese, and zinc. The study shows that temperature ranged from 16 to 32 °C, while the average pH value was 6.25. Almost all of the trace elements showed an extremely high value of electrical conductivity (average: 5713 μS/cm) and significantly high total dissolved solids (average: 3063 mg/L). The average value of chloride was 797 mg/L, which is dramatically higher than standard values. In regard to trace element concentration, almost all thermal springs were heavily contaminated with arsenic and it was recorded as 100 μg/L in the eastern part of central Bamyan. The average amounts of barium, copper, iron, manganese, and zinc were 4.14, 6.05, 1.90, 1.76, and 0.74 mg/L, respectively. In conclusion, the water of the thermal springs of central Bamyan are not suitable for human consumption because of the significant amount of trace elements as well as the high-water quality index value. Using these springs for drinking and irrigation purposes has been deemed inappropriate.

**Keywords:** thermal spring; trace element; arsenic; Bamyan; central Afghanistan; contamination



## 1. Introduction

Thermal springs are natural phenomena [1] and geological features which can be found in "active or semi active volcanic areas" [2] or observed in and around tectonic zones [1]. Generally, thermal springs have advantages and disadvantages for human and livestock use [3]. They act as the surface expression of underlying hydrothermal systems [2] and can be used for hydrothermal energy production. However, thermal water often presents anomalously high background values of different potentially toxic elements, which can present severe risks for human consumption. Due to lack of clean water resources, some consume it for drinking water [4,5], but most of the water is affected by high concentrations of arsenic [6] and other trace elements. Arsenic is also released through the thermal springs into the environment and surface/groundwater [4,7]. Contamination of groundwater through trace elements is a substantial concern for environmental health scientists since this water normally is required for human and animal consumption [8].

Research has shown that trace elements can be derived in groundwater from natural and anthropogenic sources [8].

Natural trace elements can cause health issues, even in low natural concentrations [9–13]. At the same time, there could be different local acclimation and adaption to the natural environment to existing trace concentrations. As [9] reported in a notable study, organisms in Norway adapted to relatively high trace elements such as Pb, Cu, Zn, and Mn, which could be toxic for the identical organisms in Poland. Another example is from Austria, where trace element counts for As (Arsenic) are 6–20 times higher than in other countries. However, As is bonded well naturally in As-geochemical deposits without creating health problems [9]. Proper field investigations and geomapping are very important before taking any action, including any clean-up or remediation activities [9]. Water–rock interactions are vital for thermal water chemistry studies and developing geothermal systems worldwide [14–22]. Determining the origin of thermal fluids is one the main tasks of these investigations [14].

Water resources, including thermal springs, have not been assessed in detail in Afghanistan to date, even though a range of thermal springs exist in different parts of the country. It will be helpful to carry out research on water–rock interactions and the origins of thermal water, similar to the hydrochemical studies developed by [15], in Afghanistan. Relatively recent research has focused on Afghanistan's river basins [23], creating an atlas of water for the country [24], a detailed analysis of hydrogeology of the Kabul Basin [25], evaluating groundwater availability in the Kabul Basin [26], establishing a conceptual model of groundwater resources in the Kabul Basin [27], and studying the inventory of groundwater resources in the Kabul Basin [28]. Furthermore, a preliminary study of geothermal energy including thermal springs around the country as well as thermal springs of Bamyan province was conducted by Saba et al. in 2004 [29]. This study revealed that active geothermal systems, which indicate thermal springs, are generally located in the main axis areas of the Hindu Kush running along the Herat faults. Utilization of thermal springs in Afghanistan might have begun with the settlement of the first people in the Hindu Kush valleys, [29] but since 1940, a number of thermal springs in Herat (Obe and Safed Koh), Balkh (Aabe Garm), and Orezgan provinces were developed for therapeutic purposes [29]. In 1964, an attempt was made by Soviet geologists who cooperated with the Geological Survey of Afghanistan (GSA) to conduct a systematic study on thermal waters around the country for their potential mineral content. Their study explored a number of hot springs in Kalu Valley (Ghorband, Shina, Dare-e Soof, and Istalef), some of which are observation points of the current study. [29]. It was found that thermal springs in Afghanistan are mainly bicarbonate, chloride, sulfate, and sodium chloride type [29,30]. It is important to note that a similar study of the anomalously high concentration of metals in thermal water was conducted in the Thriasion Plan (NW Attica, Greece) [31] and it showed that the total dissolved solids (TDS) ranged from 535 to 13567 mg/L, $Cl^-$: 17.7–7269.7 mg/L, $SO_4^{2-}$: 8.2–782 mg/L, and $NO_3^-$: 5.7–293 mg/L. However, in central Bamyan thermal springs, the amount of $Cl^-$ ranged from 49 to 1630 mg/L, $SO_4^{2-}$: 64–1430 mg/L, and $NO_3^-$: 1.3–21 mg/L.

Since knowledge of thermal water chemistry is still limited in Afghanistan, this work aimed to fill this knowledge gap. This aim was reached by (1) determining the contamination rate and protection strategies; (2) determining the arsenic and some trace element amounts in the thermal springs; (3) and assessing the vulnerability and health risks caused by arsenic and essential trace elements such as barium, copper, iron, manganese, and zinc.

## 2. Materials and Methods

### 2.1. Study Area

2.1.1. Location and Climate Condition

Bamyan is located in the center of Afghanistan (Figure 1) within latitude 33°48′ N to 35°28′ N and longitude 66°24′ E to 68°14′ E [32]. The specific study area is situated in the central highland of Bamyan province between two main mountainous ranges: the Hindu Kush Mountains in the north and the Baba Mountain range in the south. Geographically, it is placed within longitude 67°20′ E to 68°10′ E and latitude 34°30′ S to 35°10′ N (Figure 1), and topographic elevation is around 2520 m above sea level. Kahmard and Saighan districts are located in the northern part, the Baba Mountain range in the south, the Shaikh Ali and Surkh Pars districts of Parwan province in the east, and the Yakawlang district in the western part of study area (Figure 1).

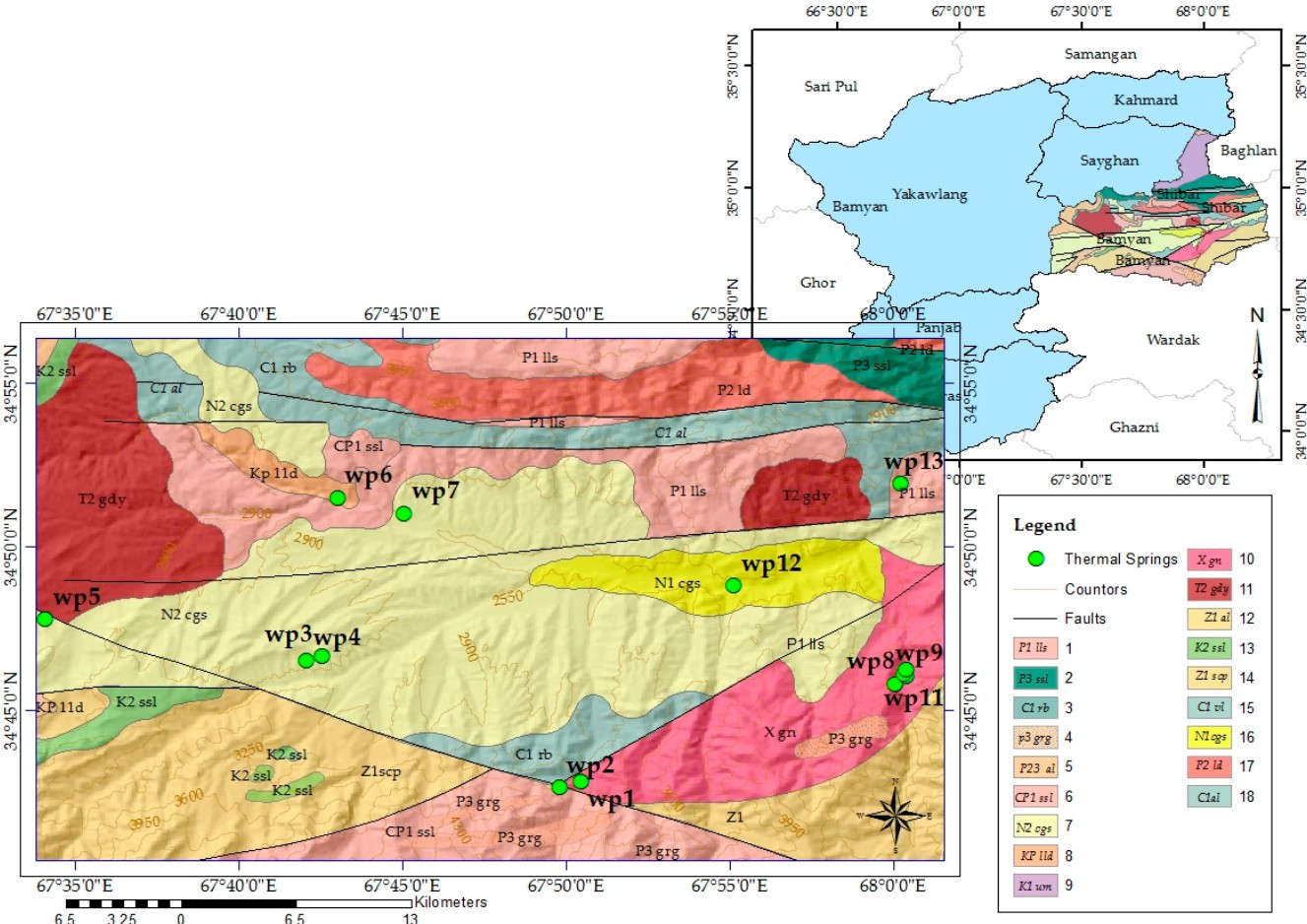

**Figure 1.** Geological map which isolated the geological setting of the study area from the entire Bamyan province. This map was created by the geographical information system (ArcGIS 10.2). In the legend: 1. Sandstone and siltstone (Early Permian), 2. Sandstone and siltstone (Oligocene), 3. Lava (Carboniferous), 4. Granite and granodiorite (Oligocene), 5. Andesite lava (Oligocene), 6. Siltstone and sandstone (Early Permian), 7. Conglomerate and sandstone (Pliocene), 8. Limestone and dolomite (Late Permian), 9. Granodiorite and granite (Early Triassic), 10. Gneisses (Paleoproterozoic), 11. Granodiorite and granosyenite (Late Triassic), 12. Metavolcanic andesite lava (Early Neoproterozoic), 13. Sandstone and siltstone lava (Late Neoproterozoic), 14. Schist and rhyolite (Early Neoproterozoic), 15. Rhyolite and basalt (Early Carboniferous), 16. Conglomerate and sandstone (Miocene), 17. Limestone and dolomite (Paleocene), 18. Andesite Lava (Early Carboniferous).

The climate of Bamyan varies significantly throughout the region from arid to semiarid, with warm summers and cold winters. During cold winters, the average temperature is −5 °C, while sometimes it can decrease to −20 °C [33]. The average recorded precipitation is 165 mm, which mainly occurs as snow [34].

### 2.1.2. Geological Setting

The geology of Bamyan valley, similar to many mountain chains around the world, consists of crystalline or metamorphic bedrock with different rock formations from the Paleoproterozoic to Quaternary periods [30]. Metamorphic and igneous rocks mainly exist in most parts of the southern hills and the Baba Mountain ridge. At higher elevations, granite to gabbro, schist, slate quartzite, and gneiss are major rock units in the southern ridges that are especially exposed [35].

Glacial and alluvial deposits of snow–glacier melting processes during the Quaternary period created hollows/cirques, huge moraines, and alluvial fans/plains that provide the major water reservoirs for Bamyan valley [30].

The northwestern Bamyan ridges (Figure 1) primarily have sedimentary rock formations comprised mostly of limestone and carbonates with an intercalation of basic to intermediate volcanic. Their ages vary from Carboniferous, to Permian, and to Cretaceous. Granitic batholiths of the northern Hindu Kush (Triassic–Jurassic) intrude the strata several times [35].

A deep east–west regional fault through Bamyan valley cuts all rock formations and channels into deep hydrothermal brines. Related hot springs and brines are widespread alongside the fault, especially in the eastern (Paymory Area) and western (Azhdar Area) sides of the city of Kabul (Figure 1).

### 2.1.3. Hydrological Setting

Bamyan is a mountainous region and the central part of this province is surrounded with two main mountains called the Baba Mountain, which is located at the southern part and prolongs from east to west, and the Hindu Kush Mountain, located at the northern part of central Bamyan and which prolongs almost parallel to the Baba Mountain. Central Bamyan is a wide valley located between the Hindu Kush Mountain range (north) and the Baba Mountain range (south). Many valleys and subvalleys connect these mountainous regions to the main Bamyan valley.

From a hydrological point of view, the main river of Bamyan flows from the main Bamyan valley (Figure 2). Almost all the southern valleys are wet during the year because the water from the rivers that flow through these valleys originates from Baba Mountain's glaciers and snow at high elevations and natural lakes. The main rivers which originate from Baba Mountain include the Foladi River, which has several subrivers: the Dokani River, the Darra Sadat River, the Darra Ahangaran River, and the Kalu River (Figure 2). Comparatively, the Hindu Kush Mountain provides less water than the Baba Mountain. Therefore, the majority of the valleys located in the northern part are dry except for some small flows which have water for a limited time during the year.

There are a variety of thermal hot and cold springs throughout central Bamyan, especially in the south, such as the Foladi, Dokani, Sadat, Agangaran, and Kalu valleys (Table 1). Further, some cold springs are situated in the Bamyan main valley, which supply the urban water systems. In some parts, these springs are used for irrigation and drinking.

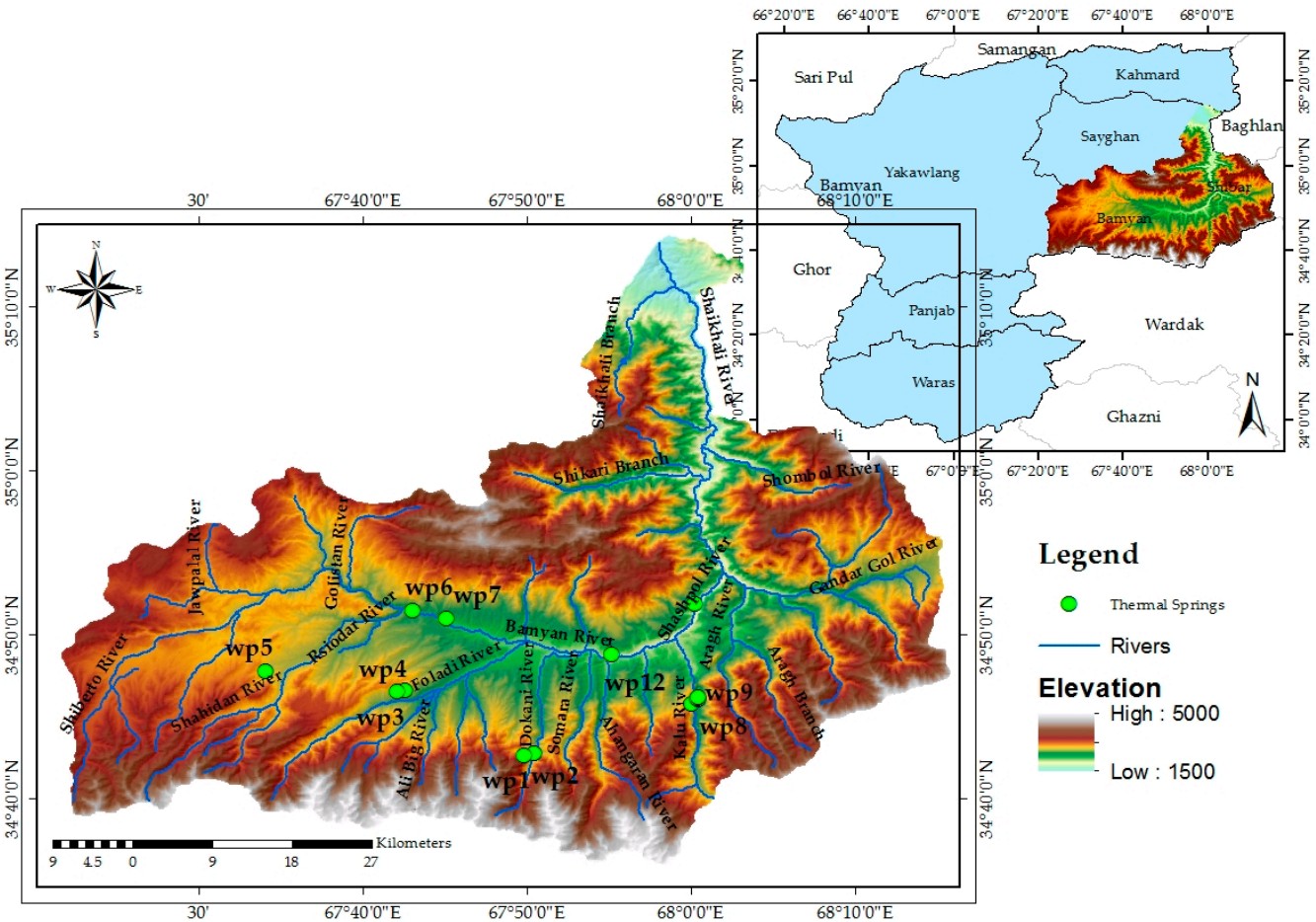

**Figure 2.** Hydrological map of study area. This map was created by the geographical information system (ArcGIS 10.2).

**Table 1.** Location of observation points (thermal springs).

| Symbol | Location | Description of Location | Coordinates (WGS1984) | | Elevation (m) |
| | | | Longitude | Latitude | |
|--------|----------|-------------------------|-----------|----------|---------------|
| wp$_1$ | Darra-e-Dokani1 | Mountainous region, located at the end of Dokani valley close to Baba Mountain ridge | 67.84 | 34.71 | 2939 |
| wp$_2$ | Darra-e-Dokani 2 | Mountainous region, located at the end of Dokani valley close to Baba Mountain ridge | 67.83 | 34.71 | 2945 |
| wp$_3$ | Syalayag Foladi1 | Located at the end and right side of Foladi valley, near Baba Mountain ridge, southwestern part of Bamyan City | 67.709 | 34.777 | 3530 |
| wp$_4$ | Syalayag Foladi 2 | Located at the end and right side of Floadi valley, near Baba Mountain ridge, southwestern part of Bamyan City | 67.701 | 34.775 | 3537 |

**Table 1.** *Cont.*

| Symbol | Location | Description of Location | Coordinates (WGS1984) | | Elevation (m) |
|---|---|---|---|---|---|
| | | | Longitude | Latitude | |
| wp$_5$ | Azdar Shahidan | Located in the Shahidan area, left side of Bamyan–Yakawlang main road with travertine and limestone sediments | 67.57 | 34.80 | 3068 |
| wp$_6$ | Khoja Ali | Located near the southern entrance gate of Bamyan City, at contact of sedimentary and igneous rocks | 67.72 | 34.86 | 2635 |
| wp$_7$ | Asidar Bamyan | Located at the end of Azhdar valley, south of Bamyan City with travertine and limestone | 67.75 | 34.85 | 2729 |
| wp$_8$ | Paymory Kalo1 | Located at the right side of Kabul–Bamyan highway, middle point of Kalu valley, eastern part of Bamyan City | 67.05 | 34.79 | 2505 |
| wp$_9$ | Paymory Kalo2 | Located at the lift side of Kabul–Bamyan highway, middle point of Kalu valley, eastern part of Bamyan City | 68.007 | 34.767 | 2504 |
| wp$_{10}$ | Paymory Kalo 3 | Located at the lift side of Kabul–Bamyan highway, middle point of Kalu valley, eastern part of Bamyan City | 68.006 | 34.768 | 2502 |
| wp$_{11}$ | Paymory Kalo 4 | Located at the lift side of Kabul–Bamyan highway, middle point of Kalu valley, eastern part of Bamyan City | 68.007 | 34.77 | 2502 |
| wp$_{12}$ | Dahana Ahangaran | Located at the beginning of Ahangaran valley, northeastern part of Bamyan City | 67.92 | 34.81 | 2431 |
| wp$_{13}$ | Shash Pul | Located at Shash Pol area, norteastern part of Bamyan City | 67.45 | 34.51 | 2361 |

*2.2. Water Sampling and Field Parameter Measurements*

The geographical location and elevation of every observation point (sampling point) were recorded by a handheld global positioning system (GPS) and the results were plotted in the maps (Figures 1 and 2) by the use of the WGS1984 geographic coordinate system. Further, in autumn (October–December 2018), water samples were collected from every study site into a 1000 mL sterile and dried glass bottles for the purpose of analyzing the essential trace elements and arsenic. A cool bag equipped with ice bags was used to keep the water samples stable, to prevent any possible chemical characteristic changes of the water samples, and to keep the water temperature at 4 °C until the laboratory analysis could be completed. During the sampling, some physicochemical parameters, including the pH, temperature (°C), electrical conductivity (mS/cm), and total dissolved solids (TDS) (mg/L), were measured directly in situ using a portable multimeter DR 2800 Spectrophotometer (manufacturer Hatch, Colorado, USA) with two probes for pH and EC.

*2.3. Water Analysis Methods*

All targeted essential trace elements and arsenic were analyzed at Green Tech water laboratory in Kabul city. Arsenic was analyzed with an EZ Arsenic Test Laboratory Kit. These kits indicate the linear range of As analysis, which were previously immobilized with the detector as 10–250 µg/L of arsenic. The amount of barium was measured by HACH model 8014, copper with HACH 8026, iron with HACH 8008, manganese with

HACH 8149, and zinc with HACH 8009 (Table 2). DR 3900 Laboratory Spectrophotometer for water analysis (manufacturer Hatch, Colorado, USA) was applied. All these methods of HACH for the purpose of water and wastewater analysis have been reviewed, approved, and accepted by the US Environmental Protection Agency (USEPA) to monitor regulatory requirements. For greater accuracy and to ensure the accuracy of the lab chemical analysis, a duplicate analysis was conducted and the physicochemical parameters were measured again in the lab and compared with the field values.

**Table 2.** Summary of methods utilized in laboratory analysis.

| Physical Tests | | | | |
|---|---|---|---|---|
| **Tests** | **Unit** | **Methods** | **\* MCL as per USEPA** | **\* MCL as per WHO** |
| pH | ( ) | HACH #8156 | 6.5–8.5 | 6.5–8.5 |
| Color | (CoPt) | HACH #8125 | 15 | 15 |
| Conductivity | (µS/cm) | HACH #8160 | - | - |
| Resistivity | (Ω.cm) | HACH-HQ Direct measurement | - | - |
| Salinity | (%) | HACH-HQ Direct measurement | - | - |
| Turbidity | (NTU) | HACH #8237 | 5 | 5 |
| TDS | (mg/L) | HACH-HQ Direct measurement | 500 | 1000 |
| Chemical Tests | | | | |
| Arsenic | (µg/L) | EZ Arsenic Test Kit | 50 | 10 |
| Barium | (mg/L) | HACH #8014 | 2 | 0.7 |
| Chloride | (mg/L) | HACH #8206 | 250 | 250 |
| Cyanide | (mg/L) | HACH# 8027 | 0.2 | 0.1 |
| Copper | (mg/L) | HACH #8026 | 1.3 | 2 |
| Hardness | (mg/Las CaCO$_3$) | HACH #8329 | 500 | 500 |
| Iron | (mg/L) | HACH #8008 | 0.3 | 0.3 |
| Manganese | (mg/L) | HACH #8149 | 0.05 | 0.1 |
| Nitrite | (mg/L) | HACH #8507 | 1 | 3 |
| Nitrate | (mg/L) | HACH #8039 | 10 | 10 |
| Phosphate | (mg/L) | HACH #8048 | - | - |
| Sulfate | (mg/L) | HACH #8051 | 250 | 400 |
| Zinc | (mg/L) | HACH #8009 | 5 | 3 |

\* MCL: Maximum contamination level for drinking purposes. TDS: Total dissolved solids. - No limit recommended.

*2.4. Water Type Determination*

The water quality index (*WQI*) is an index reflecting the composite impacts of parameters to determine the water type and assess the health risk rate of the thermal springs.

WHO guidelines were used for the *WQI* calculations. The number of parameters were selected, then the relative weight (*Wi*) of the selected parameters was calculated by this equation [36]:

$$W_i = \frac{w_i}{\sum_{i=1}^{n} w_i} \tag{1}$$

where $W_i$ means the relative weight, $w_i$ shows the weight of every parameter, and $i$ is equal to the number of parameters.

After that, the quality rating scale ($q_i$) was determined for every parameter [36]:

$$q_i = \left(\frac{C_i}{S_i}\right) \times 100 \tag{2}$$

where $q_i$ shows the quality rating according of *i*th parameter concentration, and $n$ is a chemical parameter number, $C_i$ is the concentration of every parameter of each water sample (mg/L), and $S_i$ is the WHO standard value of the parameter.

Afterward, the sub index (*SI$_i$*) of each parameter and *WQI* was calculated as by applying these equations [36]:

$$SI_i = W_i \times q_i \tag{3}$$

$$WQI = \sum SI_{i-n} \tag{4}$$

Based on *WQI*, water has an excellent quality if the *WQI* ranges from 0 to 25. Water is considered good quality if the *WQI* range is from 25 to 50. Water of poor quality has a *WQI* range from 50 to 75. Water is very poor quality if the *WQI* ranges from 75 to 100. Water that is unsuitable for human consumption has a *WQI* number greater than 100 [36].

## 3. Results

### 3.1. Physical Parameters of Thermal Springs

The physical parameters of thermal springs are shown in Table 3. The temperature values ranged from 16 °C in Azhdar spring ($wp_7$) to 32 °C in the Paymory (Figure 1) system of thermal springs ($wp_8$–$wp_{11}$), with an average of 25.4 °C. The pH ranged from 6.39 to 7.24 (average: 6.57), which shows all the thermal springs are slightly acidic. The amount of electrical conductivity varied substantially and ranged from 719 to 9150 µS/cm (average: 5713 µS/cm), which is due to the high number of ions (cations and anions) existing in the thermal springs. Almost all of the studied thermal springs except the Khaja Ali ($wp_6$: 3 CoPt) and Azhdar ($wp_7$: 14 CoPt) thermal springs were colorful and this varied considerably between 3 and 133 CoPt (average: 74.5 CoPt). Measurement of resistivity ranged from 109.3 to 1392 Ω.cm. In regard to turbidity, almost all the thermal springs had extremely high turbidity and the value of this parameter changed from 0.21 to 57 NTU (average: 37.4 NTU). The $wp_6$ in Khaja Ali thermal spring was the only study point which showed a lower TDS value (351 mg/L) than the WHO value (1000 mg/L). However, all the other springs contained extremely high TDS which ranged from 1471 to 4960 mg/L (Table 3).

**Table 3.** Physical properties of thermal springs.

| Study Points | Temperature | pH | Color | Conductivity | Resistivity | Turbidity | TDS |
|:---:|:---:|:---:|:---:|:---:|:---:|:---:|:---:|
| | (°C) | (* SU) | (CoPt) | (µS/cm) | (Ω.cm) | (NTU) | (mg/L) |
| $wp_1$ | 25 | 6.68 | 27 | 3630 | 276 | 46 | 1876 |
| $wp_2$ | 25 | 6.8 | 30 | 3720 | 260 | 40 | 1923 |
| $wp_3$ | 25 | 6.7 | 43 | 3123 | 290 | 50 | 1526 |
| $wp_4$ | 24 | 6.5 | 46 | 3000 | 305 | 52 | 1534 |
| $wp_5$ | 24 | 6.43 | 51 | 2900 | 349 | 57 | 1471 |
| $wp_6$ | 23 | 7.24 | 3 | 719 | 1392 | 0.21 | 351 |
| $wp_7$ | 16 | 6.46 | 14 | 6590 | 151.7 | 0.44 | 3530 |
| $wp_8$ | 32 | 6.41 | 120 | 8310 | 120.4 | 41.6 | 4490 |
| $wp_9$ | 31 | 6.39 | 125 | 8354 | 121.1 | 42.5 | 4502 |
| $wp_{10}$ | 31 | 6.44 | 129 | 8363 | 125 | 43.3 | 4512 |
| $wp_{11}$ | 32 | 6.56 | 117 | 8213 | 118.9 | 40.5 | 4456 |
| $wp_{12}$ | 20 | 6.39 | 133 | 9150 | 109.3 | 34 | 4960 |
| $wp_{13}$ | 23 | 6.4 | 130 | 8205 | 134 | 38.6 | 4687 |

* Standard Unit.

### 3.2. Hydrochemistry of Thermal Springs

3.2.1. Major Ions

The results of the hydrochemistry analysis of the thermal springs are given in Table 4. The chloride ($Cl^-$) amount in these thermal springs changed from 49 mg/L in $wp_6$ to 1630 mg/L in $wp_{12}$ (Figure 3f) with an average of 795 mg/L, and all the thermal springs, except for Khaja Ali thermal spring ($wp_6$), contained concentrations of chloride that are significantly higher than acceptable USEPA and WHO values (250 mg/L). The amount of cyanide in the thermal springs ranged from 0.004 to 0.022 mg/L (average: 0.014 mg/L). The hardness rate of the thermal spring water, as assessed as $CaCO_3$, was found to be in the range of 284 to 2804 mg/L (average: 1855 mg/L). The result of thermal spring hardness compared to USEPA and WHO (2011) values (Table 4) illustrates that only water of the Khaja Ali thermal spring ($wp_6$) is soft, while all the others are extremely hard. This high

value of hardness in thermal springs of central Bamyan is due to the widespread existence of sandstone and limestone in this area and their easy dissolution into the thermal springs. The amounts of nitrite and nitrate changed from 0.06 to 1.15 mg/L (average: 0.49 mg/L) and 1.3 to 21 mg/L (average: 8.93 mg/L), respectively. The amount of phosphate fluctuated from 0.08 to 0.65 mg/L, and the values of sulfate ranged from 64 to 1430 mg/L (average: 861.7 mg/L). The amount of sulfate in the thermal springs were potentially high in comparison with the values of the USEPA and WHO standards (Table 4).

**Table 4.** Hydrochemistry and trace elements of thermal springs.

| Symbol | Chloride | Cyanide | Hardness (as CaCO$_3$) | Nitrite | Nitrate | Phosphate | Sulfate | Arsenic (As) | Barium (Ba) | Copper (Cu) | Iron (Fe) | Manganese (Mn) | Zinc (Zn) |
|---|---|---|---|---|---|---|---|---|---|---|---|---|---|
| | (mg/L) | | | | | | | (µg/L) | | (mg/L) | | | |
| Wp$_1$ | 365 | 0.01 | 1260 | 0.013 | 2.5 | 0.35 | 488 | 0 | 2 | 6.84 | 0.27 | 0.97 | 0.56 |
| wp$_2$ | 378 | 0.02 | 1287 | 0.02 | 3.1 | 0.65 | 502 | 0 | 2 | 6.65 | 0.25 | 0.92 | 0.63 |
| wp$_3$ | 283 | 0.01 | 1011 | 0.02 | 2.7 | 0.15 | 310 | 10 | 2 | 7.3 | 1.1 | 0.72 | 0.55 |
| wp$_4$ | 272 | 0.01 | 994 | 0.02 | 2.82 | 0.13 | 298 | 10 | 2 | 7.8 | 1.13 | 0.79 | 0.61 |
| wp$_5$ | 262 | 0.009 | 982 | 0.02 | 3.1 | 0.19 | 340 | 10 | 2 | 9.02 | 1.8 | 0.88 | 0.5 |
| wp$_6$ | 49 | 0.004 | 284 | 0.01 | 1.8 | 0.12 | 64 | 0 | 1 | 0.11 | 0.08 | 0.017 | 0.23 |
| wp$_7$ | 590 | 0.006 | 2340 | 0.006 | 1.3 | 0.08 | 870 | 0 | 4 | 5.14 | 0.1 | 2.7 | 0.95 |
| wp$_8$ | 1280 | 0.022 | 2615 | 1.11 | 17 | 0.48 | 1370 | 100 | 6 | 6.04 | 3.49 | 2.88 | 0.82 |
| wp$_9$ | 1297 | 0.02 | 2722 | 1.13 | 19 | 0.52 | 1392 | 100 | 6.2 | 6.12 | 3.55 | 2.94 | 0.89 |
| wp$_{10}$ | 1303 | 0.02 | 2781 | 1.15 | 21 | 0.61 | 1401 | 100 | 6.5 | 6.27 | 3.89 | 3.01 | 0.94 |
| wp$_{11}$ | 1286 | 0.021 | 2804 | 1.12 | 19.4 | 0.55 | 1362 | 10 | 6.1 | 5.77 | 3.25 | 2.67 | 0.76 |
| wp$_{12}$ | 1630 | 0.013 | 2490 | 0.79 | 9.8 | 0.23 | 1430 | 0 | 7 | 5.66 | 2.11 | 1.9 | 1.18 |
| wp$_{13}$ | 1346 | 0.02 | 2549 | 0.97 | 12.6 | 0.34 | 1375 | 0 | 7 | 5.9 | 3.73 | 2.54 | 0.94 |
| USEPA | 250 | 0.2 | 500 | 1 | 10 | - | 250 | 10 | 0.7 | 1–2 | 0.3 | 0.1 | 3 |
| WHO | 250 | 0.1 | 500 | 3 | 10 | 0.70 | 400 | 10 | 0.7 | 1–2 | 0.3 | 0.1 | 3 |

- Not given.

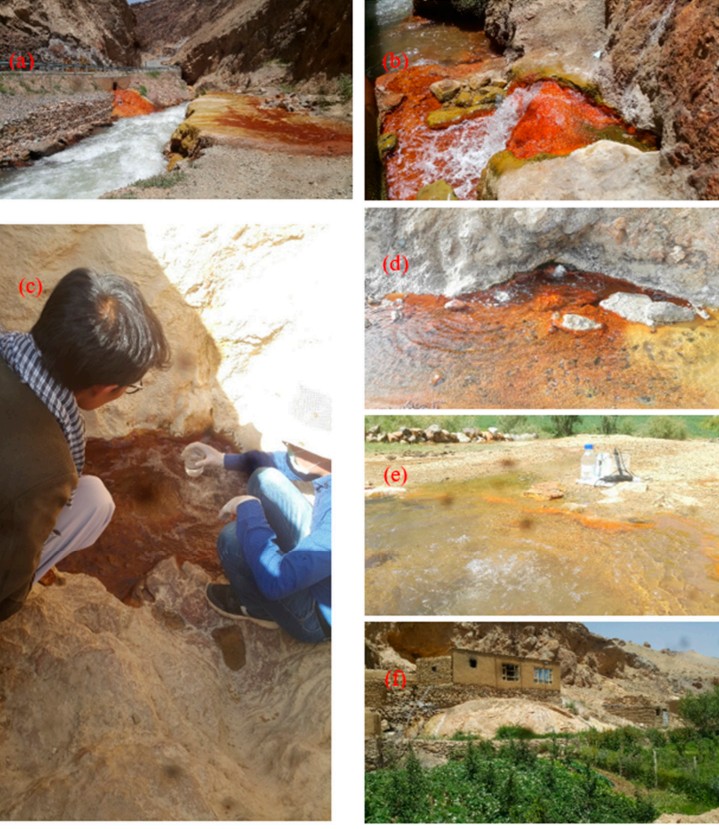

**Figure 3.** Photographs of some thermal springs: (**a**) range of thermal springs in Kalu valley, (**b**) Kalu main thermal spring, (**c**) Shahidan thermal spring, (**d**) Shash pul thermal spring, (**e**) Dokani first thermal spring, and (**f**) Darra Ahangaran thermal spring.

### 3.2.2. Trace Element Concentrations

Five trace elements, including arsenic, were assessed in the thermal springs and the results are given in Table 4. Just over half of the thermal springs contained arsenic. The amount of arsenic in the first group, located in the northwestern part of the study area, was 10 µg/L, which is equal to the maximum value of arsenic suggested by the WHO (Table 4); this included the wp$_3$, wp$_4$, and wp$_5$ (Figure 3c) study points (Figure 1). However, the value of arsenic in the second group of thermal springs, situated in the eastern part of the study area (Figure 1) and included wp$_8$, wp$_9$, and wp$_{10}$, were recorded at 100 µg/L. This amount is high compared with the accepted WHO standard amount. The high amount of arsenic in the second part of the thermal springs is potentially related to the geological singularities of the spring locations. The thermal spring range in the Kalu valley (including wp$_8$–wp$_{11}$) flows through Paleoproterozoic-geological-age gneisses bedrock, which are the oldest rocks in Bamyan valley and partly covered by sandstone. This is the reason why the thermal spring contains arsenic and trace elements. The second range of thermal springs with a concentration of 10 µg/L of arsenic (including wp$_3$, wp$_4$, and wp$_5$) flows through Pliocene-geological-age conglomerate and sandstone-based rocks, which cover the older bedrock. The high arsenic concentration in the thermal spring originates from these bedrocks.

The amount of barium fluctuated significantly and ranged from 1 to 7 mg/L, with an average of 4.14 mg/L (Figures 4 and 5), which is significantly higher than 0.7 mg/L, which is suggested in the WHO standard (Figure 6). Copper and total iron are present in the range of 0.11–9.02 mg/L (average: 6.05 mg/L) and 0.08–3.89 mg/L (average: 1.90 mg/L), respectively (Figures 4 and 5).

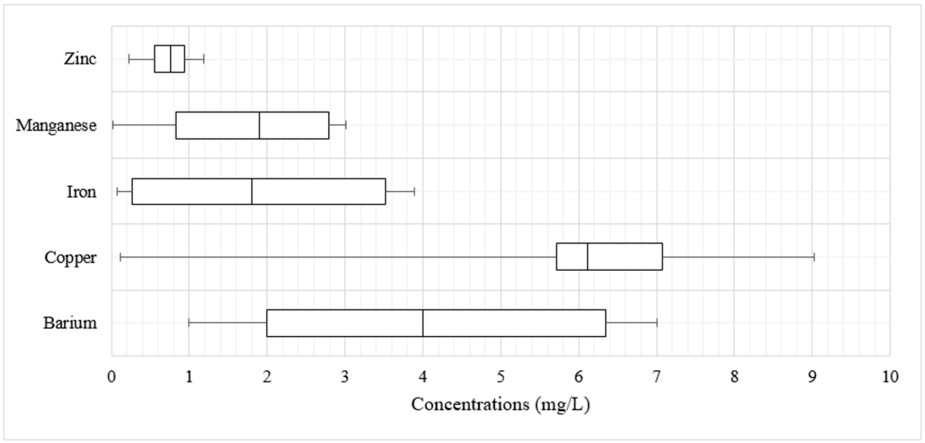

**Figure 4.** Boxplot of trace elements in thermal springs.

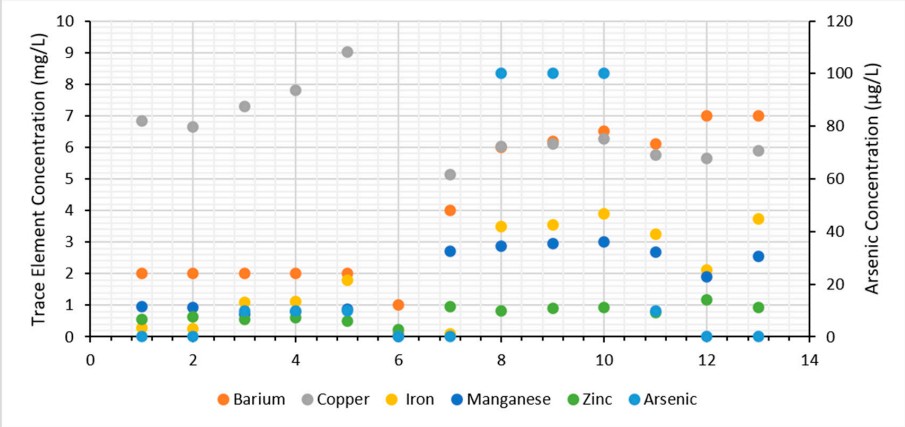

**Figure 5.** Concentrations of trace elements in thermal springs.

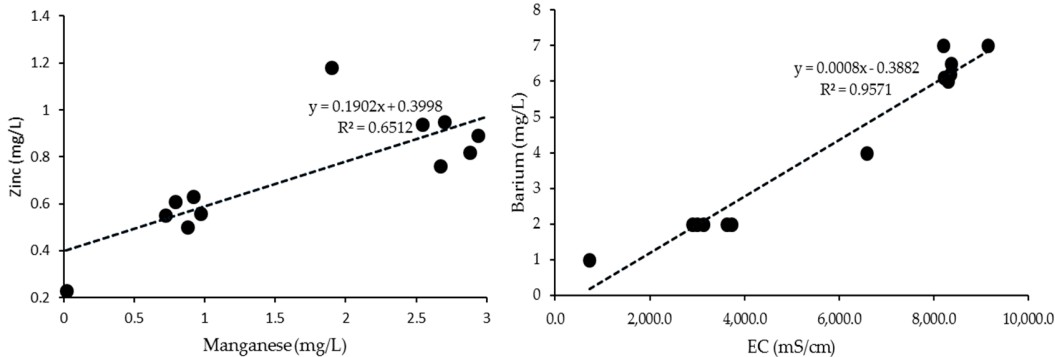

**Figure 6.** The strongest correlation between some parameters.

The concentration of manganese varied from 0.017 to 3.01 mg/L, with an average value of 1.76 mg/L, whereas zinc ranged from 0.23 to 1.18 mg/L (average: 0.74 mg/L) (see Figure 4).

*3.3. Correlations*

The study of the correlation coefficient between two variables is a way to determine the sufficiency of the first variable for the prediction of the second one [37]. This coefficient is utilized for the assessment of the correlation between the variables if the dependent variable is just influenced through the independent variable and vice versa [37].

The selection of variables was based on the physicochemical properties and the measured trace elements' importance with related correlation and determination of the variables' dependency for the trace elements (see Table 5). The results show a strong correlation ($r > 0.9$) and a good correlation ($r = <0.9$ to $>0.5$). The strongest correlations are recognized between electrical conductivity and total dissolved solids ($r = 0.999$), barium ($r = 0.978$) (see Figure 6), manganese ($r = 0932$), and zinc ($r = 0.916$), as well as between TDS and manganese ($r = 0.930$) and zinc ($r = 0.913$). Further, barium is strongly correlated with manganese ($r = 0.892$) and zinc ($r = 0.871$) and between manganese and zinc ($r = 0.807$) (see Table 5 and Figure 7). The constituents originate from the same source, and the thermal springs have near equal origins. The trace elements were from the same geological settings, which are located at the thermal springs' pathway. These strong correlations between parameters represent the variables' dependency. Investigations of the geological setting and mineral composition of the bedrock are planned for future studies.

**Table 5.** Correlation between studied parameters of thermal springs.

|           | PH    | EC   | Turbidity | TDS  | Arsenic | Barium | Copper | Iron | Manganese | Zinc |
|-----------|-------|------|-----------|------|---------|--------|--------|------|-----------|------|
| PH        | 1     |      |           |      |         |        |        |      |           |      |
| EC        | −0.73 | 1    |           |      |         |        |        |      |           |      |
| Turbidity | −0.46 | 0.06 | 1         |      |         |        |        |      |           |      |
| TDS       | −0.72 | 0.99 | 0.05      | 1    |         |        |        |      |           |      |
| Arsenic   | −0.38 | 0.49 | 0.22      | 0.48 | 1       |        |        |      |           |      |
| Barium    | −0.68 | 0.97 | 0.06      | 0.98 | 0.49    | 1      |        |      |           |      |
| Copper    | −0.66 | 0.11 | 0.84      | 0.09 | 0.08    | 0.03   | 1      |      |           |      |
| Iron      | −0.63 | 0.77 | 0.39      | 0.78 | 0.67    | 0.84   | 0.19   | 1    |           |      |
| Manganese | −0.71 | 0.93 | −0.01     | 0.93 | 0.61    | 0.89   | 0.10   | 0.76 | 1         |      |
| Zinc      | −0.78 | 0.92 | 0.03      | 0.91 | 0.31    | 0.87   | 0.21   | 0.56 | 0.81      | 1    |

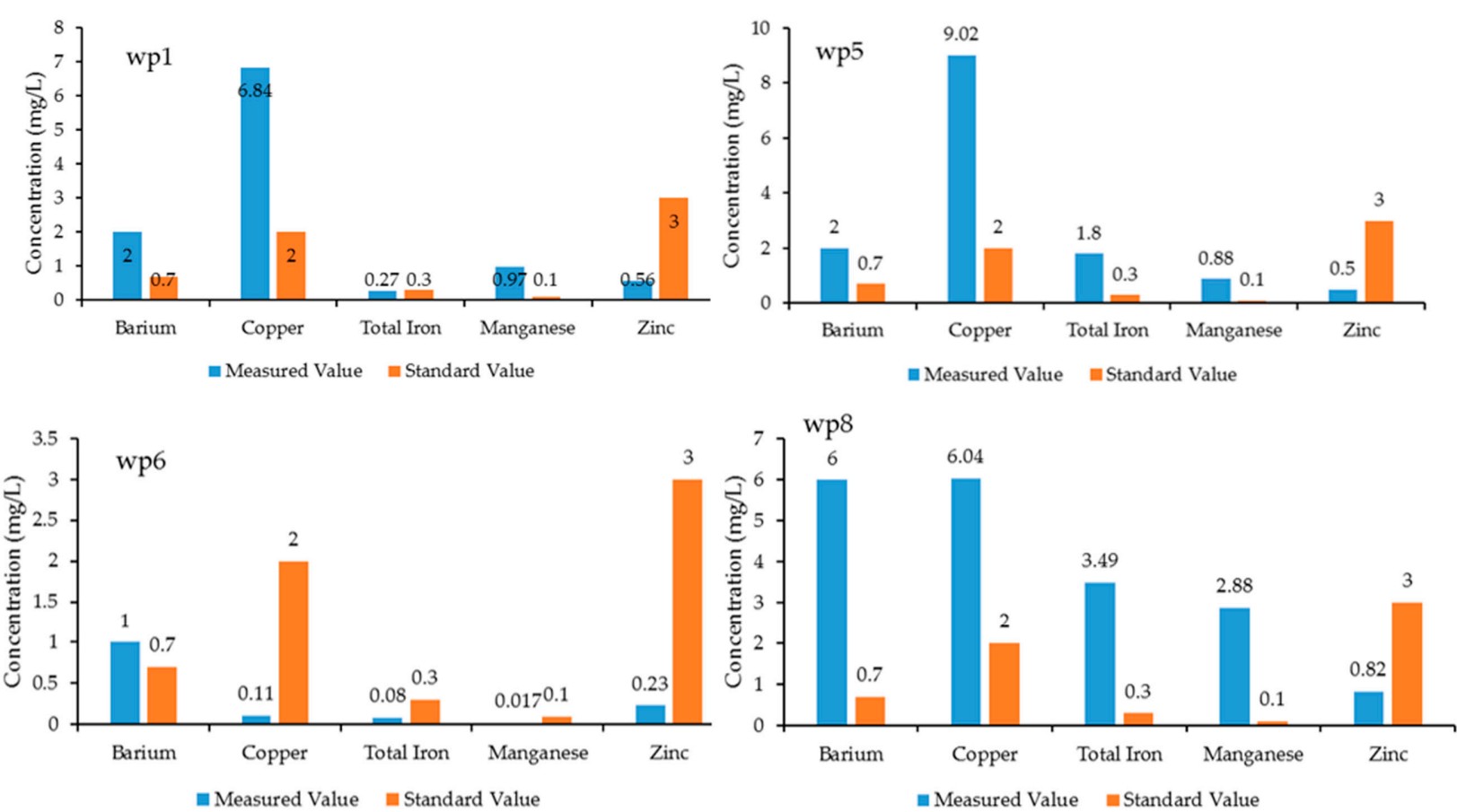

**Figure 7.** Compression of measured values of trace elements with standard WHO values.

### 3.4. Water Type and Health Risk Assessment

In many parts of the study area, the thermal springs flow into the surface water, including rivers, and this water is directly used for drinking or irrigation purposes. The result outputs based on WQI were compiled in Figure 8. All the thermal springs water samples have high numbers, compared to the WQI = 100, maximum standard value for potable water. Only the Khaja Ali thermal spring (wp6 with WQI = 17) is suitable for drinking purpose.

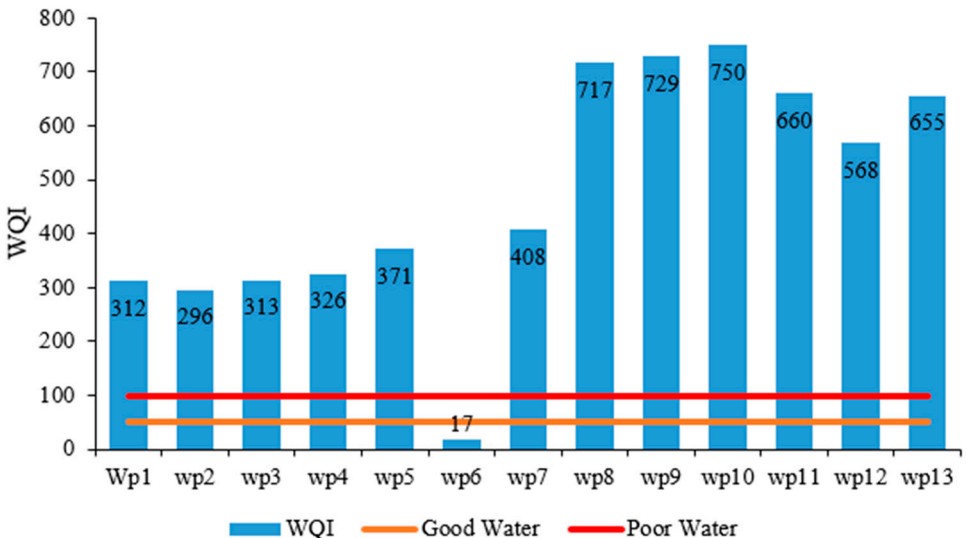

**Figure 8.** Determination of water type based on water quality index.

### 4. Discussion

Determination of the natural situation of the groundwater chemistry and water–rock interactions is one the of main aims of physicochemical studies [38–40]. Here, physical parameters such as temperature, pH, color, conductivity, resistivity, salinity, turbidity, and TDS were measured in the study area. Almost all of the water of the thermal springs was from a yellowish to brownish color. The values of almost all the water at the study points were higher than the WHO value (15 CoPt). However, the pH levels of all the thermal springs were in the standard range (6.5–8.5) and varied from 6.39 to 7.24. The amount of total dissolved solids (TDS) was extremely high (Max. = 4960 mg/L, Min. = 351 mg/L, Average = 3063 mg/L) compared with the WHO standard (1000 mg/L).

Arsenic concentrations in thermal water present values of one to three orders of magnitude higher compared with the cold groundwater [38]. Arsenic is introduced into the water via an accumulation of related deposits such as iron, sulfates, oxides, and hydroxides [41,42]. It can originate naturally and artificially: natural arsenic contamination can occur in different geological and climatic conditions [6]. The maximum amount of arsenic in the thermal springs of this study area was 100 μg/L, which is significantly higher than the maximum value of WHO (10 μg/L). In addition to arsenic, some other trace elements such as barium, copper, iron, manganese, and zinc were assessed in the thermal springs. The average amount of barium was 4.14 mg/L, which is much higher than the 0.7 mg/L maximum value of WHO. Concentrations of copper, iron, manganese, and zinc averaged at 6.05, 1.90, 1.76, and 0.74 mg/L respectively. Thus, the thermal water sources of central Bamyan are significantly contaminated with trace elements.

According to other research [37,43], water is considered to be of excellent quality if the *WQI* ranges from 0 to 25, water is considered to be good quality if the *WQI* has a range from 25 to 50, water has poor quality if the *WQI* is from 50 to 75, water has very poor quality if *WQI* ranges from 75 to 100, and water is unsuitable for drinking with a *WQI* greater than

100. As mentioned above, all the thermal springs in this study, except ($wp_6$), are in very bad condition because the *WQI* values are much higher than 100.

Overall, the thermal springs of central Bamyan that were studied are heavily contaminated with arsenic and some trace elements. Thus, this water poses a significant health risk if it is used for drinking and irrigation.

## 5. Conclusions

Thermal springs are natural phenomena with high concentrations of metals, including arsenic, as essential trace elements. Hence, it is imperative to determine the amount of these trace elements in thermal springs in order to protect public health. Originally, the majority of the thermal springs in central Bamyan were formed due to national and local tectonic activities except for the Khaja Ali thermal spring, which mainly originated from volcanic activities. This study found that some thermal springs located northeast of Bamyan City (in Kalu valley) have anomalously high concentrations of arsenic, copper, and barium. The amount of zinc in the thermal springs is significantly higher than acceptable standards. Electrical conductivity has a strong correlation with total dissolved solids, barium, manganese, and zinc, as well as barium with manganese. Based on the calculated values of the water quality index, water from all of these thermal springs are unsuitable for drinking except for the Khaja Ali thermal spring, which is of excellent quality. Thus, these sources of water should not be used for the purpose of drinking and irrigation.

**Author Contributions:** Conceptualization, methodology, H.A.J. and J.S.; software, H.A.J. and M.A.M.; validation, H.A.M., H.A.J., formal analysis, H.A.J. and J.S.; investigation, H.A.J., H.A.M., M.A.M.; data curation, H.A.J., J.S. and H.A.M.; writing-original draft preparation, H.A.J.; writing-review and editing, H.A.J., J.S., H.A.M. and M.A.M.; supervision, J.S., H.A.M., M.A.M. and H.A.J.; project administration, H.A.J., J.S., H.A.M., M.A.M. All authors have read and agreed to the published version of the manuscript.

**Funding:** The data used in this current paper was part of a research project which was supported by the University of Central Asia (UCA), the Institute of Mountain Research (MSRi); a research grant from IDRC (Canada); and the US National Academies of Sciences, Engineering, and Medicine (NAS) USAID Partnerships for Enhanced Engagement in Research (PEER) cycle 5 project Grant Award Number AID-OAA-A-11-00012.

**Institutional Review Board Statement:** Not applicable.

**Informed Consent Statement:** Not applicable.

**Acknowledgments:** This study was supported by the University of Central Asia (UCA), the Institute of Mountain Research (MSRi) and through a research grant from IDRC (Canada). The writers are thankful to the UCA staff, especially Aziz Ali Khan, for supporting and assisting us during the study. Some tests were conducted in the laboratory of the Soil Science Department, Faculty of Agriculture and we are grateful for their support. Further, the publication was supported by the Department of Civil and Environmental Engineering, School of Engineering and Digital Sciences, the Environment and Resource Efficiency Cluster (EREC) of Nazarbayev University, Kazakhstan, and the US National Academies of Sciences, Engineering, and Medicine (NAS) USAID Partnerships for Enhanced Engagement in Research (PEER) cycle 5 project Grant Award Number AID-OAA-A-11-00012. Authors also acknowledge the anonymous reviewers whose valuable suggestions helped us in improving the quality of this paper.

**Conflicts of Interest:** The authors declare no conflict of interest.

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
