# Peer review of "Essential Trace Elements and Arsenic in Thermal Springs, Afghanistan"

_water, doi:10.3390/w13020134_

Round 1

Reviewer 1 Report

The paper "Essential trace elements and arsenic in thermal springs of central Bamyan, Afghanistan" reports some interesting results about geochemical anomalies possibly related with the tectonic setting of different geothermal springs in Afghanistan. This paper present different interesting  results which could worth publication, but still need a more detailed analysis and especially a deeper discussion of the results with major comparison with present literature in the field. Moreover, a careful revision of English usage along the whole text should be performed, there different errors and misspellings.

Starting from the introduction, I suggest to rewrite the first sentence of the paper as "Thermal springs are natural hydrogeological features which are highly affect from local volcanisms or tectonic." Then, I suggest to rephrase the sentence in lines 36-37 as "However, thermal water often present anomalously high background values of different potentially toxic elements, which can present severe risks for human consumption"

Then, I suggest to shorten the paragraph in lines 43-53. It is in fact known that different case studies regarding geochemical background values are diffused worldwide. Authors should also add some literature regarding regional studies (see for example Reimann & Garret 2005, https://doi.org/10.1016/j.scitotenv.2005.01.047).
Then I would focus more on thermal systems and their relationships with geology and tectonics (see for example Vengosh et al 2002, https://doi.org/10.1016/S0883-2927(01)00062-2 or Binda et al. 2020, https://doi.org/10.3390/min10121058 ).

Moreover, after presenting the case studies in Afghanistan, in the final sentence of the introduction, authors should better stress the fact that knowledge on chemical quality of thermal water in Iran is limited, highlighting then the novelty of the paper.

Moving to methods, in line 93 I suggested to remove the sentence "is very complicated". This sentence is in fact too generic, and often all mountain chains presenting crystalline or metamorphic bedrock present complicated geological settings.

In figure 1 the labels for springs not readeable. I also suggest to zoom in a separate panel, to better indicate WP9-WP11 points.

In figure 2 there is a double lable for rivers, and the misspelling for "elevation" to correct.

In line 126, I suggest to rewrite "From a hydrological point of view,".

In table 1, I suggest to maintain the sampling point number or the symbol only, since the repetition of similar sampling points can be confounding. Moreover, I suggest to indicate the coordinate format. Is it WGS84?

In figure 3, Please add the sampling point label in the caption.

Then in line 155, please add more details regarding analytical protocols. Are these HACH methods following other protocols (e.g. US EPA)? What are the detection limits for the protocols? Are these protocols portable  field kits or laboratory protocols? These details can help for a better interpretation of results.

Then, in lines 164-166, the use of GIS for the site map creation is not a real statistical analysis. I suggest to add this sentence in figure 1 and 2 caption, instead of methods.
Authors should instead add some more details regarding coorelation analysis. I also suggest to test some other multivariate analysis technique to better highlight correlation among chemical variables (e.g. PCA , Liang et al 2018 https://doi.org/10.3390/ijerph15122752 ;or cluster analysis, Binda et al. 2020, https://doi.org/10.1007/s10653-019-00405-4 ). Moreover, in line 164, the authors state "GIS 10.3". Which software do they mean? ArcGIS, QGIS?

Moving to results, in table 3 "pH" instead of "PH". Moreover, it is bit redundant to explain conductivity, salinity and TDS values, which are 3 different unit to measure the same water property.

In figure 4, what do the authors mean with "total iron"? All the other elements should be considered total. I suggest to remove total.

In figure 5, I suggest to remove the line connecting different As measurements, which is not significative. Moreover, in my opinion the graph will have a better appearance with the same style (bar or points) for all elements. Also, correct "Arsenic" label for axis title.

In lines 228-234, authors should explain the selection of variables made for correlation analysis. Why do they select trace elements and some physico-chemical properties? Maybe other major ionjs can give interesting correlations (see for example Battistel et al. 2020 https://doi.org/10.1016/j.gexplo.2020.106534). Moreover, it is too simplistic to say that the correlation among concentration means the same source for diferent elements. Are there any known minerals in the area presenting the same elements reported? Are the dissolution rates or the saturation indexes known?

In figure 6, please change "Zinc" label style.

In table 6 caption, please change "paramteres" in "parameters".

In figure 7, as they are these graphs are hard to read. I suggest for example to add the limit as lines or symbols, in order to avoid overlap of different bars, similarly to figure 8.

In line 243, authors should describe water quality index calculation in the "methods" section.

Line 252: "or" intead of "0r".

Line 253: "except for" instead of "except".

Also, table 6 is redundant with figure 8. I suggest to remove it.

Generally, the discussion section need more detailed investigation and a more detailed comparison with other published literature. For example, sentence in lines 263-264 is too generic. Then, in line 271 Authors should carefully use the term "pollution". In this case, thermal waters present anomalously high concentration of metals, which are not related with pollution. And once again, in lines 276-279 authors are too generic in their description. Authors should try to evaluate the trends with other elements to better understand the geochemically-derived arsenic in the springs (see e.g. Ghoreyshinia et al, 2020,
https://doi.org/10.1016/j.apgeochem.2020.104629).

Therefore, I suggest that after these revisions the paper can be considered for publication in Water.

Author Response

Dear Reviewer,
we have updated our manuscript based on your recommendations.
Please review the attached file.
Thank you very much for your time.

Reviewer 2 Report

I have reviewed the work entitled Essential trace elements and arsenic in thermal springs of central Bamyan, Afghanistan by Jawadi et al., and my considerations are as follows.

General comments

The authors present research about essential trace elements in thermal springs of central Bamyan, in Afghanistan. For this purpose, they have evaluated the concentrations of trace elements in relation to standard limits for WHO.

The manuscript is written with a logical sequence. Structurally is adequate: all the key elements are present (abstract, introduction, methodology, results and discussion, and summary and conclusions). However, the present manuscript could be improved. The discussion of results and conclusions detract from the effort made, as the first part of the article shows. A thorough review of this last part will be very beneficial and can certainly enrich the manuscript.

Specific comments 

Main text

Line (38)

If the source of arsenic is natural, perhaps ‘contaminated’ is not the best word. It could be changed by ‘affected by high concentrations of As’, for example.

General

Please, check points (two points in line 106), and double spaces (i.e. line 177).

Water sampling and field parameter measurements

Eh values are very important parameter in this context. Perhaps the author could fill this lack doing some considerations with the values of Fe and Mn (see bibliography about that). It looks like it could be a reducing environment.

Table 4

Column of Barium must be checked. In general, all data could be revised to provide the data with the same significant digits.

Figure 5

It would be interesting to take into account the high values of As in some samples and relate them to geological singularities. Very useful for the Discussion of the results.

Water type and health risk assessment.

The WQI should be more and better explained. The bibliographic references, their usefulness, how it has been applied, etc. It would help the reader to understand the results it offers.

An index that can also be useful and perhaps contrasted, since it deals with trace elements and their relationship with the thresholds established for drinking water, is Hydrogeotoxicity, HGT (Giménez-Forcada, E.; Vega-Alegre, M.; Timón-Sánchez, S. 2017. Hydrogeotoxicity from Arsenic and Uranium in the southern Duero Basin, Spain. Journal of Geochemical Exploration, 183B: 197-205.)

Discussion

The discussion of the results can be improved.

The first paragraph (lines 258-263) could be deleted. The text corresponding to lines 263-270, can be more and better developed. The text on lines 271-275 looks like it would be better in the Introduction. Text 276-291 seems redundant and not conclusive.

Conclusions

The conclusions also deserve a review.

Author Response

(The authors gave the same response as above.)

Round 2

Reviewer 1 Report

I read the revisions performed by the authors on the paper "Essential trace elements and arsenic in thermal springs of central Bamyan, Afghanistan". The authors clearly improved the paper quality. Nonetheless, I still found some misleading sentences and few typos. I suggest to the authors to carefully check for English grammar. Please find a list of minor revisions which need to be checked by the authors.

Line 11: "highly affected" instead of "highly affect"

Line 27: use comma instead of "-"

Line 46: I suggest to rephrase the sentence as "As reported in a notable research, organisms..." then, in the following line, please indicate which are the trace elements showing anomalous values.

Line 74: "and it showed that"

In the last paragraph of the introduction (lines 77-82), I suggest to move the last sentence at the beginning, starting with "Since the knowledge on thermal water chemistry is still limited in Afghanistan, this work aims to fill this knowledge gap. This aim will be reached through (1) the determination of the contamination rates and protection strategies attempt; (2) determination of arsenic and some trace elements in the thermal springs; (3) and the vulnerability assessment..."

Line 167-168: I do not understand the meaning of this sentence. Is it to indicate the linear range of As analysis kit? It should be rewritten.

Lines 242-244: Authors should remove "moves through" and maintain "flows through" only. Then, in the following sentence, authors should better explain which kind of rocks are the older bedrock, and I think they are covered by sandstones.

Table 4: please use always the capital "L" for concentration units.

Line 267: "physico-chemical" instead of "physic-chemical"

Lines 301-302: Authors should rephrase as "Arsenic concentrations in thermal water present values of one to three orders of magnitude higher compared to the cold groundwater"

Author Response

Dear Reviewer,

the manuscript was updated.

Please review the attached file.

Thank you very much for your time.

Happy New Year !

Reviewer 2 Report

The article review is correct.

Author Response

Dear Reviewer,

Thank you very much for your time.

Happy New Year !